# Domain memory effect in the organic ferroics

Zhong-Xia Wang [1,2✉], Xiao-Gang Chen[2], Xian-Jiang Song[2], Yu-Ling Zeng[2], Peng-Fei Li [2],
Yuan-Yuan Tang [2], Wei-Qiang Liao[2] & Ren-Gen Xiong [2✉]

Shape memory alloys have been used extensively in actuators, couplings, medical guide wires, and smart devices, because of their unique shape memory effect and superelasticity triggered by the reversible martensitic phase transformations. For ferroic materials, however, almost no memory effects have been found for their ferroic domains after reversible phase transformations. Here, we present a pair of single-component organic enantiomorphic ferroelectric/ferroelastic crystals, (R)- and (S)-N-3,5-di-tert-butylsalicylidene-1-(1-naphthyl) ethylamine SA-NPh-(R) and SA-NPh-(S). It is notable that not only can their ferroic domain patterns disappear and reappear during reversible thermodynamic phase transformations, but they can also disappear and reappear during reversible light-driven phase transformations induced by enol–keto photoisomerization, both of which are from $P1$ to $P2_1$ polar space groups. Most importantly, the domain patterns are exactly the same in the initial and final states, demonstrating the existence of a memory effect for the ferroic domains in SA-NPh-(R) and SA-NPh-(S). As far as we are aware, the domain memory effect triggered by both thermodynamic and light-driven ferroelectric/ferroelastic phase transformations remains unexplored in ferroic materials. Thermal and optical control of domain memory effect would open up a fresh research field for smart ferroic materials.

[1] College of Chemistry and Chemical Engineering, Gannan Normal University, Ganzhou 341000, People's Republic of China. [2] Ordered Matter Science Research Center, Nanchang University, Nanchang 330031, People's Republic of China. ✉email: zhongxiawang@ncu.edu.cn; xiongrg@ncu.edu.cn

Shape memory alloys are smart materials that have a broad range of industrial and commercial applications, such as actuators, couplings, medical guide wires, and so on[1,2]. The reversible martensitic phase transformations bring about the unique shape memory effect and superelasticity in these alloys[3]. However, it is difficult for the ferroic domains to show memory effect after ferroics undergo reversible phase transformations. Ferroics, generally including ferromagnets, ferroelectrics, and ferroelastics, have at least two orientation states that could be switched by a certain external field in the low-temperature ferroic phase[4–9]. Taking ferroelectrics as an example, the transformations from ferroelectric phase to paraelectric phase and back to ferroelectric phase would result in increased and broken crystal symmetries[10–19]. Due to the influence of temperature changing rate and crystal defects, the shape and size of newly emerged ferroelectric domains often change greatly from the original ones, which can not realize the memory effect. For instance, in two ferroelectric phases of quinuclidinium perrhenate, both domain patterns gained during the cooling treatment are obviously different from those gained in the original states[20].

Temperature control is a time-consuming and energy-cost process, and the fixed temperature range partly limits their wider application fields. Light irradiation, as a non-destructive, non-contact, rapidly responsive, and remote-control means, can address the above-mentioned limitations[21–24]. Inspired by a classic scene in *Men in Black*, where the agent uses a mysterious flash to erase the memory of human beings, the instantaneous erasure and reproduction of memory might also be realized in the near future. However, the optical control of ferroic orders is still a great challenge so far, which has been long pursued as a dream treasure in the scientific community. Very recently, it has been demonstrated that light irradiation could be applied to control the polarization switching for organic ferroelectrics[25–28]. For example, our group reported single-component organic crystals (*R*)- and (*S*)-*N*-3,5-di-*tert*-butylsalicylidene-1-4-bromophenylethylamine and 3,4,5-trifluoro-*N*-(3,5-di-*tert*-butylsalicylidene)aniline, which can realize the polarization switching via a structural phase transformation triggered by photo-induced enol–keto geometrical isomerization[25,26]. Unfortunately, they suffer from very slow domain switching and/or difficult-to-recover domain structures under laser irradiation, where the shape and size of ferroelectric domains are very different in the original and annealed states.

During our systematic search, we found that Taniguchi et al. have reported the photo-triggered phase transformation with enol–keto photoisomerization in a homochiral and polar crystal, (*S*)-*N*-3,5-di-*tert*-butylsalicylidene-1-(1-naphthyl)ethylamine (SA-NPh-(*S*))[29]. The polar point group is the necessary condition of being ferroelectrics. Enlighted by this pioneering work, we have systematically studied a pair of single-component organic enantiomorphic crystals, SA-NPh-(*R*) and SA-NPh-(*S*), which show not only temperature-dependent thermodynamic ferroelectric/ferroelastic phase transformation but also light-driven ferroelectric/ferroelastic phase transformation triggered by enol–keto photoisomerization both from *P*1 to *P*2$_1$ polar space groups. Accompanied by the ferroelectric/ferroelastic phase transitions, ferroelectric/ferroelastic domains can be switched quickly within seconds and reversibly between different polarization/strain states. Moreover, their spontaneous polarization or ferroelectric domain can also be switched under opposite electric fields in all three phases (room-temperature and high-temperature phases, and photo-triggered phase). Consequently, ferroelectric/ferroelastic domains can be simultaneously controlled by three different physical channels (light, temperature, and electric field). To the best of our knowledge, the multi-channel (optical, thermal, and electronic) control of ferroelectric/ferroelastic domains realized in organic crystals is previously unknown. In addition to the

electrically switchable polarization in the ferroic phase, the native ferroelectric/ferroelastic domain patterns (State A) can completely disappear (State B) by heat or 365 nm UV irradiation and reversibly reemerge (State A') upon cooling or 488 nm visible light irradiation (Fig. 1). Most importantly, the domain patterns in the final state (State A') are exactly the same as in the original state (State A), demonstrating the presence of memory effect for the ferroic domains in SA-NPh-(*R*) and SA-NPh-(*S*). Such an intriguing domain memory effect triggered by both thermodynamic and light-driven ferroelectric/ferroelastic phase transformations has been rarely reported in ferroic materials, which provides a future direction for smart ferroic materials.

## Results and discussion

**Phase transformation.** Single crystals of SA-NPh-(*R*) and SA-NPh-(*S*) were easily prepared by evaporating their methanol solutions at room temperature. The previous report has shown the thermal and photo-induced phase transformations of SA-NPh-(*S*)[29]. At 273 K (Phase I below $T_c$), two homochiral organic compounds of SA-NPh-(*R*) and SA-NPh-(*S*) crystallize in triclinic enantiomorphic polar space group *P*1, while their racemic mixture forms a monoclinic structure with centrosymmetric space group *P*2$_1$/*n* (Table S1). Structurally, their asymmetric units in Phase I adopt a *trans*-enol molecular structure, and three C atoms in one of the *tert*-butyl groups of molecule need to be split to model a disordered state for better structural refinement results (Fig. 2 and Fig. S1). Differential scanning calorimetry (DSC) analyses reflect that both SA-NPh-(*R*) and SA-NPh-(*S*) experience thermodynamic phase transformations at around $T_c = 312$ K (*R*) and 317 K (*S*), respectively (Fig. S2a and b), however, the racemic mixture has no thermodynamic phase transformation (Fig. S3). At 323 K (Phase II above $T_c$), their structures are changed into the monoclinic polar space group *P*2$_1$ (Table S2), where thermal vibrations of atoms become more intense, showing a more disordered state (Fig. 2 and Fig. S1). Therefore, they undergo 2*F*1 type ferroelectric-paraelectric phase transformation

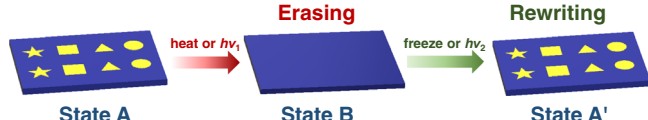

**Fig. 1 Diagram of domains changes by heat and light.** The erasing and rewriting process of ferroic domains controlled through temperature or light irradiation. The different shapes marked with yellow color denote the domains.

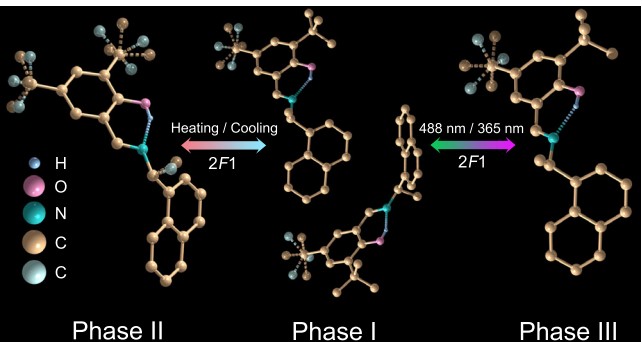

**Fig. 2 Crystal structures.** Asymmetric units of SA-NPh-(*R*) in Phase I, II, and III. The dotted lines between N and H atoms indicate O-H...N hydrogen bonding. Some H atoms were omitted for clarity. The disordered *tert*-butyl and methyl substituents are distinguished by tan and light turquoise colors.

and their thermodynamic mechanism can be attributed to the order-disorder transformation of molecules. As shown in Figs. S4–S6, the blue and red arrows represent the directions of the molecular dipole moment and the total polarization of the unit cell, respectively. Molecular dipoles adopt similar arrangements in these three different unit cells. Therefore, the total polarization of the three phases is also similar, except that the deformation of the unit cell changes significantly.

**Ferroelectric properties**. Generally, the dielectric anomaly usually accompanies the paraelectric-ferroelectric phase transformation, and thus we carried out the temperature-dependent dielectric constant ($\varepsilon$) of SA-NPh-(R). Here, we take SA-NPh-(R) as an example to discuss, due to the similar structure and properties of SA-NPh-(R) and SA-NPh-(S). Figure S2c shows the real part ($\varepsilon'$) of the dielectric constant of SA-NPh-(R). It is clear that an obvious anomaly or a sharp peak emerges around $T_c$, indicative of a proper ferroelectric transition[30,31]. We then performed the polarization−voltage (P–V) hysteresis loop measurements using the double-wave method to demonstrate the ferroelectricity of SA-NPh-(R). As shown in Fig. S2d, two opposite peaks appear in the current density–voltage (J–V) curve at room temperature, suggesting two stable states with opposite polarization. From the current accumulating, we can acquire the typical P–V hysteresis loop, indicating the presence of ferroelectricity in SA-NPh-(R). Above $T_c$, the ferroelectricity of SA-NPh-(R) in Phase II was also confirmed. In Fig. S7, a similar ferroelectric loop was obtained because of the small difference in the total polarization between Phases I and II (Figs. S4 and S5). We also measured the ferroelectric hysteresis loop in Phase III. However, there is no significant difference between the loops in Phases I and III (Fig. S8). The main reason might be attributed to the close total polarization of the two phases (Phases I and III). Moreover, the generally low conversion rate of the enol−keto isomerization in the Schiff base crystals upon photoirradiation results in less difference for the ferroelectric loops.

**Optical properties**. Photochromism in salicylideneaniline Schiff base derivates generally involves electron transfer and configuration transformation under photoexcitation. In detail, when the crystals are irradiated by UV light, the hydrogen proton transfer from the hydroxyl group to the bridge nitrogen atom is activated, and the stable enol form is transformed to the unstable intermediate state of cis-keto form. With continuous UV light irradiation, the cis-keto form will be immediately transformed to the relatively stable trans-keto form with lower steric hindrance that occurs via pedal motion in a manner similar to the thermal motion of azobenzene crystals (Fig. S9)[32,33]. Under UV irradiation of 365 nm at 298 K, SA-NPh-(R) and SA-NPh-(S) also crystallize in the same polar space group $P2_1$ as that in Phase II (Table S3), while their molecules display a disordered state similar to those in Phase I. The conversion rate of photoisomerization in the single crystal is not enough, and thus the obtained structure from X-ray determination still consists almost of enol molecules (Fig. 2 and Fig. S1), where the pure trans-keto molecules are difficult to separate. Fortunately, we can use other measurements and calculations to confirm the existence of trans-keto molecules (see below). According to the Aizu rule, both thermal and light-driven phase transformations belong to the $2F1$-type[34], which is not only a full ferroelectric one but also a full ferroelastic one, providing opportunities to realize optical control of multiple ferroic orders, especially photo-control polarization switching. It should be noted that to rule out the thermal effects of light and identify the intrinsic photoirradiation mechanism in the phase transformation, IR thermal imaging analyses were carried out

after UV or visible light irradiation. As shown in Fig. S10, the temperature of SA-NPh-(S) sample after UV or visible light irradiation is much lower than $T_c$, indicating that light irradiation is unique in our work and does not produce some level of photothermal illusion.

The HOMO (highest occupied molecular orbital) and LUMO (lowest unoccupied molecular orbital) of SA-NPh-(R) and SA-NPh-(S) molecules were calculated to get a better understanding of their electronic structure. Because the left and right-handed molecules have mirror symmetry, the corresponding HOMO and LUMO also present the relationship of mirror image, respectively (Fig. S11a). From the HOMO of their molecules, the electron density is mostly distributed on the benzene ring. Compared to the HOMO, the LUMO has a significant shift to the C–N direction. The resultant energy gap between HOMO and LUMO is 0.16925 Hartree, which corresponds to the energy of 4.43 eV, close to the result estimated from the experimental UV-vis absorption spectra (see below).

The chiral features of SA-NPh-(R) and SA-NPh-(S) were investigated by room-temperature circular dichroism (CD) spectra in the UV-vis absorption range, as shown in Fig. S11b. They show the same and strong CD signals peaked at 293, 237, and 208 nm, which are exactly mirror-image symmetry relation. It is clear that the CD signals originate from the Cotton effects of the intrinsic molecular π-conjugated absorption bands. Intriguingly, their CD spectra changed significantly as SA-NPh-(R) and SA-NPh-(S) samples were exposed to UV irradiation of 365 nm, where new broad CD peaks appeared from 400 to 550 nm. This signal change is ascribed to the great conformational transformation of the molecule after UV light irradiation, resulting in a corresponding change in the electronic transformation of chiral molecules.

Such photo-induced geometrical isomerization under UV light can bring about photochromic behaviors (Table S4). We then carried out solid-state UV-vis absorption spectra of SA-NPh-(R) and SA-NPh-(S) before and after UV light irradiation for investigating the color change of crystals (Fig. S11c). Under the ambient condition, both of them absorb light <400 nm, agreeing well with their yellow appearance. After UV light irradiation, a new absorption band appears with the wavelength range of 400-550 nm, in line with the orange color. Hence, the photochromic effect results from the emergence of new absorption bands under UV light irradiation.

Vibrational circular dichroism (VCD) measurements were performed to further investigate the chirality of SA-NPh-(R) and SA-NPh-(S) crystals. As shown in Fig. S11d, both enantiomorphic crystals have almost the same IR spectra, whereas their VCD spectra show a near-mirror relationship, demonstrating the enantiomorphic nature of SA-NPh-(R) and SA-NPh-(S) crystals. The VCD spectra of SA-NPh-(R) and SA-NPh-(S) crystals display several pairs of strong signals ($\Delta\varepsilon$) centered at 1,440, 1,398, 1,248, and 1,170 cm$^{-1}$, and some relative weak dichroic signals at 1,467, 1,360, and 1,236 cm$^{-1}$, which correspond well to the specific IR vibration peaks. When we calculated the VCD and IR spectra, we found that the strongest VCD signal at 1440 cm$^{-1}$ should result from the C*-C bond torsional vibration and the C–C stretching vibrations of the naphthalene ring framework. In comparison with the measured counterparts, the calculated IR and VCD spectra exhibit a slight peak shift, which might originate from the structural relaxation after optimization (Fig. S12). The VCD spectra were performed as well at Phases II and III, respectively. There are no significant changes in the measured wavenumber range (Fig. S13), suggesting that the chirality keeps the same under these two conditions.

The IR spectra of SA-NPh-(S) were measured for exploring the origin of the changes in molecular conformation before and after

UV light irradiation. As shown in Fig. S14, the most significant change in IR spectra after UV light irradiation is the emergence of a new absorption peak centered at 3380 cm$^{-1}$. According to the experimentally measured structure from single-crystal X-ray diffraction, several different molecular conformations, enol, *cis*-keto, and *trans*-keto forms were constructed (Fig. S15). Further frequency calculation reveals that only *trans*-keto form shows a peak at the range >3100 cm$^{-1}$ (Fig. S16). The calculated wavenumber (3407 cm$^{-1}$) at this peak originates from the free N–H stretching vibration, which is close to the measured result (3380 cm$^{-1}$). Due to the existence of N–H···O hydrogen bond, the *cis*-keto form does not have a free N–H stretching vibration mode. Hence, the molecule is most likely to be converted from enol form to *trans*-keto form. After irradiation with 365 nm UV light, the molecular dipole moment increased to 6.57 Debye as the molecule was converted from the enol form (2.14 Debye) to the *trans*-keto form (Fig. S2e). Further irradiation with 488 nm visible light can transfer the *trans*-keto form back to enol form. Such kind of molecular switching under light irradiation is rare in ferroelectric materials.

We also performed the second harmonic generation (SHG) measurements to confirm the crystal symmetry and phase transformation[35]. Figure S2f presents a detectable SHG signal at room temperature, in accordance with the noncentrosymmetric polar structure in SA-NPh-(*R*). With the increase in temperature, the SHG intensity increases gradually from Phase I to Phase II. However, under UV light irradiation of 365 nm, the SHG intensity becomes weaker in Phase III compared with that in Phase I. Two different phase transformation mechanisms result in different variation trends of SHG intensity under thermal condition and optical irradiation, respectively.

**Ferroelastic domains**. The occurrence of ferroelastic phase transformation is usually accompanied by the appearance and disappearance of ferroelastic domains. The different directions of ferroelastic domains have different birefringence characteristics under perpendicularly polarized light, and thus show bright and dark patterns[36]. Figure 3a exhibits beautiful stripe ferroelastic domains in SA-NPh-(*S*) thin film at 293 K using polarized light microscopy. These domains are very stable in Phase I. As the temperature increases into Phase II (323 K), these domains completely disappear, corresponding to the domain erasing process from the ferroelastic phase to the paraelastic one (Fig. 3b). When cooled back to Phase I (293 K), the stripe domains reappear (known as the domain rewriting process), indicating that SA-NPh-(*S*) undergoes a ferroelastic transformation and returns to the ferroelastic phase (Fig. 3c). It should be noted that the

reemerged domain patterns are the same as their initial ones. In addition to the thermally driven ferroelastic phase transformation, SA-NPh-(*S*) also displays light-driven ferroelastic phase transformation as well. After UV light irradiation of 365 nm within seconds, the stripe ferroelastic domains can also completely disappear, which corresponds to the erasing process of the domain, suggesting that Phase III is also a light-triggered paraelastic phase (Fig. 3d,e). Subsequently, upon visible light irradiation of 488 nm, the stripe birefringence pattern reappears in the thin film (known as the rewriting process), which drives a transformation from the light-triggered paraelastic phase to the light-triggered ferroelastic phase (Fig. 3f). Notably, the light-driven erasing/rewriting process also shows good reversibility of domains. As a result, the ferroelastic phase transformation can be triggered by two different physical approaches, namely, thermally and optically, while the optically triggered process not only is a rapid and remote-controlled way but also has high reversibility. Similar to SA-NPh-(*S*), SA-NPh-(*R*) also shows the thermal or optical control of ferroelastic domains, as shown in Fig. S17.

**Ferroelectric domains**. Piezoresponse force microscopy (PFM) becomes a promising method for visualizing and switching local ferroelectric polarization at the microscale, which can further confirm the ferroelectricity in SA-NPh-(*R*) and SA-NPh-(*S*)[37]. The box-in-box switched domain patterns can be written on the thin film samples of SA-NPh-(*R*) and SA-NPh-(*S*) in both Phases I and II (Figs. S18–S21), with the application of opposite voltages, offering solid proof for electrically ferroelectric polarization switching.

Next, we show that the ferroelectric domains in SA-NPh-(*S*) thin-film can achieve erasing and recovery behaviors under both temperature and light irradiation. Figure 4a shows the initial stripe domain pattern of SA-NPh-(*S*) thin film in Phase I from vertical PFM mapping. We then heated the sample to Phase II (323 K), which leads to the switch of all red domains to blue domains, representing the lowest energy configuration of lattice distortion in response to the phase transformation (Fig. 4b). Afterward, the sample was cooled to room temperature. We observed that the red domains reappear at their previous location (Fig. 4c). Furthermore, light irradiation with different wavelengths can also realize the polarization switching in SA-NPh-(*R*) and SA-NPh-(*S*). When we illuminate the same area with a UV light of 365 nm within seconds, the PFM mapping becomes a uniform contrast, corresponding to the mono-domain state (Fig. 4d). This indicates that after the irradiation of 365 nm UV light the ferroelectric domain pattern could be erased rapidly. Local PFM switching spectroscopy verifies the robust ferroelectricity in Phase III after 365 nm UV light irradiation, as demonstrated by 180° phase hysteresis and butterfly-shaped amplitude loops (Fig. S22a). After visible light irradiation of 488 nm, we found that the domain structure could completely return to its original state (Fig. 4e), showing the light-controlled rewriting process of the ferroelectric domain. The PFM switching spectroscopy was performed again to confirm the ferroelectricity of the sample after 488 nm visible light irradiation (Fig. S22b). Consequently, light irradiation has been demonstrated to present a reversible erasing and rewriting process of the domain pattern for SA-NPh-(*S*) in a fast and noncontact way.

Combined with the above analyses, the photoisomerization behavior between two ferroelectric phases of Phase I and Phase III should be responsible for this reversible light-controlled erasing and rewriting process. The photoreaction mainly occurs on the sample surface but not in the core because the light is difficult to penetrate the interior of crystals. Although the conversion ratio of the photoisomerization is not high for bulk materials, the thin

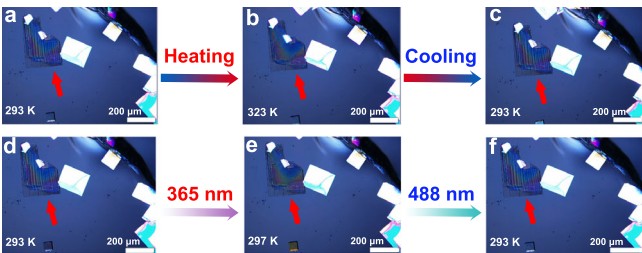

**Fig. 3 Evolution of ferroelastic domains.** The evolution of ferroelastic domains under the variation of temperature and light for SA-NPh-(*S*), the scale bar is 200 μm. The red arrows point to crystals with ferroelastic domains. **a–c** Temperature-controlled process: **a** initial state in Phase I. **b** Heating into Phase II (erasing). **c** Cooling back into Phase I (rewriting). **d–f** Light-controlled process: **d** initial state in Phase I. **e** Phase III upon UV light irradiation of 365 nm (erasing). **f** Returning into Phase I upon visible light irradiation of 488 nm (rewriting).

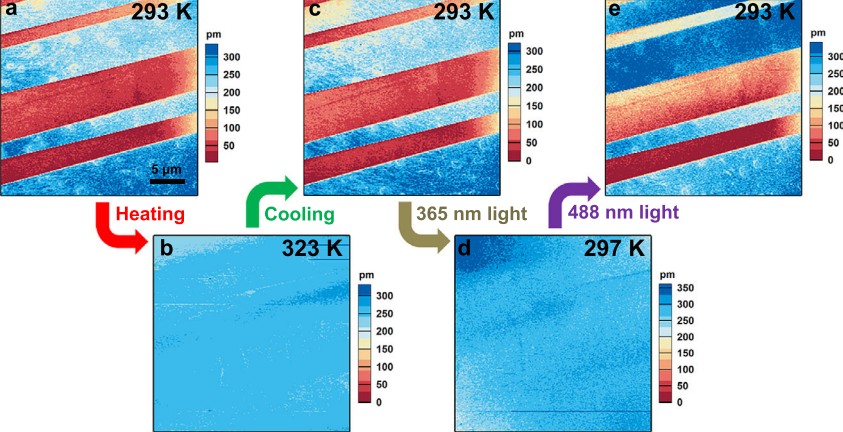

**Fig. 4 Ferroelectric domains change by heat and light.** Consecutive images showing erasure and recurrence of stripe domains in SA-NPh-(*S*) thin-film via both thermal treatment and light irradiation. The red and blue regions indicate the two different polarization-oriented states of ferroelectric domains. Domain patterns for SA-NPh-(*S*) thin-film **a** at the initial state, **b** after heating to the Phase II and **c** cooling down to the room temperature, subsequently **d** after 365 nm UV light irradiation, and **e** after 488 visible light irradiation.

film can achieve relatively high conversion efficiency under light irradiation and ferroelectric polarization reversal can also be realized in the surface micro-areas scanned by PFM. Upon UV light irradiation of 365 nm, SA-NPh-(*S*) becomes into the *trans*-keto form from the original enol form and thereby transforms into a light-triggered para-phase where the domain pattern would be erased. After visible light irradiation of 488 nm, the *trans*-keto form would return to the enol form, known as the light-triggered ferro-phase, and thus the domain patterns reemerge as the initial state. The fact that light can act directly on the molecules themselves gives them a unique advantage that the ferroelectric domain pattern could be optically controlled to disappear and reemerge, and this can be achieved in arbitrary domains without the help of device configuration or specific domain architecture.

To summarize, we have shown the reversibly thermal- and light-driven ferroelectric/ferroelastic phase transformation in a pair of single-component optically erasable/rewritable organic enantiomeric crystals SA-NPh-(*R*) and SA-NPh-(*S*). Upon UV/ visible light irradiation, they can undergo a ferroelectric/ ferroelastic structural phase transformation with an enol–keto photoisomerization, and thereby induce the optically controlled erasing and rewriting process toward ferroelectric/ferroelastic domains. The disappearing or reemerging of graphic memory is realized by the transformation of enol–keto photoisomerization between the light-triggered ferro- and para-phases via simple photoirradiation, without resorting to specific conditions and tuning complex interactions. Light control shows its unique superiority of fast response, long-range control, and non-destructive processing with relatively lower power consumption, which are promising for graphic data memory in some extreme cases such as military, deepwater, or aviation operations. This work provides an efficient material system for realizing the optically erasable/rewritable multiferroic graphic memory and should inspire further exploration for the next generation of optically controllable ferroelectric/ferroelastic devices.

## Methods

**Materials.** The general procedure for the preparation of SA-NPh-(*R*), SA-NPh-(*S*) and SA-NPh-(*Rac*) is shown in Fig. S9. To the methanol solution of 3,5-di-*tert*-butyl-2-hydroxybenzaldehyde (7.299 g, 30 mmol) in a 250 mL round bottom flask equipped with a spherical condenser, naphthylethylamine (5.137 g, 30 mmol) was added and then stirred for refluxing overnight. After completion of the reaction, the solution was slowly cooled to room temperature, filtered, and pale-yellow solid

was obtained with a good yield. The single crystals of SA-NPh-(*Rac*) were obtained by slowly evaporating the methanol solution at room temperature. The synthetic procedures of SA-NPh-(*R*) and SA-NPh-(*S*) are similar to that of SA-NPh-(*Rac*) with naphthylethylamine replaced by its homochiral compounds (*R*)-naphthy-lethylamine and (*S*)-naphthylethylamine, respectively. The phase purity of as-grown crystals are verified by the powder X-ray diffraction (PXRD), where the measured PXRD patterns of these crystals at 293 K match well with the ones simulated by their crystal data respectively (Fig. S23).

**Thin-film preparation.** The precursor solutions of these crystals were prepared by dissolving 20 mg of as-grown crystals in 500 μL of methanol solution. Then, 20 μL of precursor solution was dropped and spread on a clean indium-doped tin oxide (ITO) glass substrate (1 × 1 cm). The prepared thin films were obtained after annealing at 306 K for about 30 min. The film is about 3 μm thick.

**Measurements.** Methods of single-crystal X-ray diffraction, DSC and SHG measurements, UV-vis diffuse-reflectance spectra, CD spectra, VCD and IR measurements, PFM measurements, and related theoretical analysis were described in supporting information.

**Reporting summary.** Further information on research design is available in the Nature Research Reporting Summary linked to this article.

## Data availability

All data generated and analyzed in this study are included in the Article and its Supplementary Information, and are also available from corresponding authors upon request. The crystal structures generated in this study have been deposited in the Cambridge Crystallographic Data Centre under accession code CCDC: 2070777-2070781, 2070783-2070785, and can be obtained free of charge from the CCDC via www.ccdc.cam.ac.uk/data_request/cif.

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

## Acknowledgements

This work was supported by the National Natural Science Foundation of China (21991142, 21831004, and 92156015).

## Author contributions

Z.-X.W. synthesized the samples, wrote the manuscript, and helped conceive the study. X.-J.S. and Y.-Y.T. carried out the PFM study. X.-G.C. and Y.-L.Z. carried out the general characterizations. P.-F.L. and W.-Q.L. carried out the theoretical analysis. R.-G.X. conceived the study and wrote the manuscript with input from other authors.

## Competing interests

The authors declare no competing interests.
