## [Peer review file · Nature Communications]

REVIEWER COMMENTS

Reviewer #1 (Remarks to the Author):

Original paper on shape memory effect in ferroelectric materials.

The influence/mechanism of irradiation by light needs to be much more clear - e.g. clarify the impact of purely light-driven form of excitation rather than some level of photo-thermal heating; e.g. Figure 1 the use of “up” and “down” is vague. There is some discussion of this in the paper, e.g. the IR thermal imaging analysis (Fig S11), but it is easy to miss and a clear differentiation could be made. e.g. add T to later figures (such as the important Figure 4) to make clear the temperature has only changed during healing/cooling and not during 365nm/488nm illumination.

Figure 2 - indicate directions of polarisation for ferroelectric phases.

Figure S2c - Why is 1MHz chosen for permittivity peak at T_c . For Fig S2c(d) what is the temperature of testing in relation to T_c ? Is there are ferroelectric loop above T_c to show ferroelectric properties disappear?

“ Although the conversion ratio is not high, the photoisomerization of the sample surface should have been basically completed. “ - unclear - please rewrite.

Paper review of potential interest on light and heat in ferroelectric materials;

Mutual insight on ferroelectrics and hybrid halide perovskites: a platform for future multifunctional energy conversion R Pandey et al *Advanced Materials* 31 (43), 1807376 2019

Reviewer #2 (Remarks to the Author):

Light irradiation controlled polarization switching for organic ferroelectrics is already known including recent reports by the same authors. Photo-triggered phase transformation of SA-NPh-(S) has also been reported. This paper utilizes this for better Light irradiation controlled polarization switching for organic ferroelectrics which is an achievement but not adequate for a Nature Communication paper in terms of stringent novelty and impact criteria. The paper may be reconsidered for publication in Scientific Report.

Reviewer #3 (Remarks to the Author):

In this manuscript, the authors demonstrate interesting domain memory effects in a pair of single-component enantiomeric organic photochromic multiferroics SA-NPh-(R) and SA-NPh-(S). They simultaneously exhibit unprecedented thermal- and light-driven ferroelectric/ferroelastic phase transitions. More importantly, the ferroic domains can be reversibly controlled by three different physical channels (light, temperature, and electric field) with the perfect reproducibility and memory. This is a landmark work that thermal and optical control of domain memory effect would open up a whole new research field for smart ferroic materials, suggesting the great potential in erasable data storage, optoelectronics, and related applications. Given the manuscript is fluently written, and extensively elaborates the purpose of the research, principal results, and major conclusions, I strongly recommend its publication in Nature Communications after minor revisions.

1. The reaction time of photoisomerization in UV, CD, and IR spectroscopy tests in this article is ambiguous. The corresponding reaction time should be given to be more scientifically rigorous.
2. The authors are suggested to discuss the mechanism of photochromism in detail.
3. How about the photo-irradiation effect in P - E hysteresis curve?

Reviewer #1 (Remarks to the Author):

Original paper on shape memory effect in ferroelectric materials.

The influence/mechanism of irradiation by light needs to be much more clear - e.g. clarify the impact of purely light-driven form of excitation rather than some level of photo-thermal heating; e.g Figure 1 the use of “up” and “down” is vague. There is some discussion of this in the paper, e.g. the IR thermal imaging analysis (Fig S11), but it is easy to miss and a clear differentiation could be made. e.g. add T to later figures (such as the important Figure 4) to make clear the temperature has only changed during healing/cooling and not during 365nm/488nm illumination.

Response: We would like to express our appreciation to the reviewer for his/her helpful comments and questions, which are crucial for us to improve the quality of our manuscript. We also would like to thank him/her for the seriousness and carefulness in the reviewing process.

When different wavelengths of light act on a material, part of the energy is converted into heat, which will possibly affect the physical properties of the material. In this work, SA-NPh Schiff base not only has thermal triggered structural phase transition but also absorbs light to exhibit light-driven structural phase transition with prominent photochromism. To rule out the thermal effects of light and identify the intrinsic photoirradiation effect, IR thermal imaging analyses were carried out after UV or visible light irradiation (Fig. S11). The results demonstrate that after irradiation with 365 nm UV light, the temperature of the sample changes about 4 °C, and the maximum temperature after radiation is much lower than the T_c . When the sample is irradiated with 488 nm visible light, the temperature experiences inconspicuous changes. Therefore, we believe that light irradiation is unique in our work and does not produce some level of photothermal illusion. Following the reviewer's suggestions, we have made the new description in the part of **Optical properties** to clarify the impact of a purely light-driven mechanism in the revised version. Moreover, we have also modified the figures and the captions (including Figs. 1, 3, 4, S11 and S12) to make clear the temperature has only changed during healing/cooling and not during 365 nm/488 nm irradiation in the revised version. For example, we have added the temperature of testing in Figs. 3, 4 and S12. The use of “up” and “down” in the caption of Fig. 1 is vague, which would make a certain confusion, and thus we have redrawn Figure 1 and modified the caption for clarity. Some discussions for Fig. 1 have also been added to the main text. We have also revised the caption of Fig. S11 for clear differentiation between the images.

Figure 1 in the previous version.

Figure 3 in the previous version.

Figure 4 in the previous version.

Figure S11 in the previous version. Infrared thermal imaging analyses of SA-NPh-(S).

Figure S12 in the previous version.

The corresponding changes in the revised version:

It should be noted that to rule out the thermal effects of light and identify the intrinsic photoirradiation mechanism in the phase transformation, IR thermal imaging analyses were carried out after UV or visible light irradiation. As shown in Fig. S10, the temperature of SA-NPh-(S) sample after UV or visible light irradiation is much lower than T_c , indicating that light irradiation is unique in our work and does not produce

some level of photothermal illusion.

In addition to the electrically switchable polarization in the ferroic phase, the native ferroelectric/ferroelastic domain patterns (State A) can completely disappear (State B) by heat or 365 nm UV irradiation and reversibly reemerge (State A') upon cooling or 488 nm visible light irradiation (Fig. 1). Most importantly, the domain patterns in the final state (State A') are exactly the same as in the original state (State A), demonstrating the presence of memory effect for the ferroic domains in SA-NPh-(R) and (S).

Figure 1 | Diagram of domains (represented by different shapes) change by temperature and light. The erasing and rewriting process of ferroic domains controlled through temperature or light irradiation.

Figure 3 | Evolution of ferroelastic domains. The evolution of ferroelastic domains (red arrow) under the variation of temperature and light for SA-NPh-(S), the scale bar is 200 μm. a-c) Temperature-controlled process: a) initial state in Phase I. b) heating into Phase II (erasing). c) cooling back into Phase I (rewriting). d-f) Light-controlled process: d) initial state in Phase I. e) Phase III upon UV light irradiation of 365 nm (erasing). f) returning into Phase I upon visible light irradiation of 488 nm (rewriting).

Figure 4 | Ferroelectric domains change by heat and light. Consecutive images showing erasure and recurrence of stripe domains in SA-NPh-(*S*) thin-film via both thermal treatment and light irradiation. Domain patterns for SA-NPh-(*S*) thin-film a) at the initial state, b) after heating to the Phase II and c) cooling down to the room temperature, subsequently d) after 365 nm UV light irradiation, and e) after 488 visible light irradiation.

Figure S10. Infrared thermal imaging analyses of SA-NPh-(*S*). a) sample in enol form irradiated under 365 nm UV light. b) sample in *trans*-keto form irradiated under 488

nm visible light. c) temperature variation with the 365 nm UV light on and off. d) temperature variation with the 488 nm visible light on and off.

Figure S17. The evolution of ferroelastic domains under the variation of temperature and light for SA-NPh-(R), the scale bar is 200 μm.

Figure 2 - indicate directions of polarisation for ferroelectric phases.

Response: We thank the reviewer for the good advice. For clarity, Figure 2 was plotted to only show the structural changes by heat and light. Following the reviewer's suggestion, we have added the schematic diagrams of polarization directions in the respective three polar phases and the corresponding descriptions in the revised version, as shown in Figs. S4-S6.

The corresponding changes in the revised version:

As shown in Figs. S4-S6, the blue and red arrows represent the directions of the molecular dipole moment and the total polarization of the unit cell, respectively. Molecular dipoles adopt similar arrangements in these three different unit cells. Therefore, the total polarization of three phases is also similar, except that the deformation of the unit cell changes significantly.

Figure S4. Schematic diagram of the molecular dipole (blue arrows) direction and the total polarization (red arrow) direction of the unit cell in Phase I with space group $P1$.

Figure S5. Schematic diagram of the molecular dipole (blue arrows) direction and the total polarization (red arrow) direction of the unit cell in Phase II with space group $P2_1$.

Figure S6. Schematic diagram of the molecular dipole (blue arrows) direction and the total polarization (red arrow) direction of the unit cell in Phase III with space group $P2_1$.

Figure S2c - Why is 1MHz chosen for permittivity peak at T_c . For Fig S2c(d) what is the temperature of testing in relation to T_c ? Is there are ferroelectric loop above T_c to show ferroelectric properties disappear?

Response: We thank the reviewer for the instructive comments. Typically, the frequency range for dielectric measurements is 500 Hz to 1 MHz. Due to the interference of space charge and leakage current at low frequency, its dielectric response is usually not as convincing as that at high frequency. On the other hand, the most sensitive frequency range where the orientation polarization of molecular dipoles in the molecular ferroelectric system is also around 1 MHz. Therefore, we chose dielectric data at 1 MHz to illustrate the dielectric anomaly in ferroelectric phase transitions.

Figure S2d shows the ferroelectric hysteresis loop at room temperature, which has been already indicated in the main text. For clarity, we have added the illustration in the caption of Figure S2d in the revised version. At Phase II above T_c , their structures also crystallize in the polar space group. We actually have used PFM to confirm the ferroelectricity at Phase II. The box-in-box switched domain patterns can be written on the thin-film samples of SA-NPh-(*R*) and SA-NPh-(*S*) at Phases II (Figs. S19 and S21), with the application of opposite voltages, offering solid proof for electrically ferroelectric polarization switching. In this revision, we have also added the

ferroelectric loop of Phase II to clearly reveal its ferroelectricity.

The corresponding changes in the revised version:

Above T_c , the ferroelectricity of SA-NPh-(R) in Phase II was also confirmed. In Fig. S7, a similar ferroelectric loop was obtained because of the small difference in the total polarization between Phases I and II (Figs. S4 and S5).

Figure S2. Phase transformation and ferroelectric properties. a,b) DSC curves of SA-NPh-(R) and (S). c) The real part (ϵ') of the dielectric permittivity of SA-NPh-(R) measured on the single crystal along the crystallographic c -axis. d) Current density-voltage (J - V) curve and the corresponding polarization-voltage (P - V) hysteresis loop of SA-NPh-(R) at room temperature. The current peaks are ascribed to the ferroelectric switching. e) Schematic diagram of molecular conformation transformation of SA-NPh-(S). f) Temperature-dependent SHG response for

SA-NPh-(R). Inset: Comparison of SHG intensity in Phase I and Phase III.

Figure S7. Polarization–voltage (P – V) hysteresis loop of SA-NPh-(R) at 323 K at phase II.

“ Although the conversion ratio is not high, the photoisomerization of the sample surface should have been basically completed. “ - unclear - please rewrite.

Response: We thank the reviewer for the kind comment. Photochromic crystals generally exhibit only a small proportion of molecules that are transformed to their photoirradiation state and the yields of photoproducts are often low. This happens in the case of photochromic SA-NPh crystals in our work and other photochromic salicylideneaniline Schiff base derivatives since the photoreaction mainly occurs on the sample surface but not in the core because the UV light is difficult to penetrate the interior of crystals (*Chem. Soc. Rev.* **2004**, *33*, 579; *J. Am. Chem. Soc.* **1999**, *121*, 5809; *Commun. Chem.* **2019**, *2*, 19). PFM is a surface technique based on scanning probe microscopy. An a.c. voltage is applied through a conductive tip, causing piezoresponse of the ferroelectric sample due to the converse piezoelectric effect. According to the amplitude and phase of the piezo-response, polarization distribution and domain structures can be derived. For this reason, we carried out PFM measurements to characterize ferroelectric polarization reversal in microscopic scale. Because the thin film can achieve relatively high conversion efficiency under light excitation compared to bulk crystals, we can observe obvious ferroelectric polarization changes using PFM. Following the reviewer’s suggestion, we have rewritten the sentence “Although the conversion ratio is not high, the photoisomerization of the sample surface should have been basically completed.” into “The photoreaction mainly occurs on the sample surface but not in the core because the light is difficult to penetrate the interior of crystals. Although the conversion ratio of the photoisomerization is not high for bulk materials, the thin film can achieve

relatively high conversion efficiency under light excitation and ferroelectric polarization reversal can also be realized in the surface micro-areas scanned by PFM.”

Papereview of potential interest on light and heat in ferroelectric materials;
Mutual insight on ferroelectrics and hybrid halide perovskites: a platform for future multifunctional energy conversion R Pandey et al *Advanced Materials* 31 (43), 1807376 2019

Response: We thank the reviewer for the good suggestion. We have carefully read the article mentioned by the reviewer. The article reviews the perovskite materials for future multifunctional energy conversion and storage devices. The authors also mention that multiple physical effects (light, heat, electric, and magnetic fields) synergy in one material can realize an improved efficiency. This article motivates us to develop new multifunctional ferroelectric materials. We have cited this article in the proper place in the revised version.

The corresponding changes in the revised version:

24. Pandey R, *et al.* Mutual Insight on Ferroelectrics and Hybrid Halide Perovskites: A Platform for Future Multifunctional Energy Conversion. *Adv Mater* **31**, 1807376 (2019).

Reviewer #2 (Remarks to the Author):

Light irradiation controlled polarization switching for organic ferroelectrics is already known including recent reports by the same authors. Photo-triggered phase transformation of SA-NPh-(S) has also been reported. This paper utilizes this for better Light irradiation controlled polarization switching for organic ferroelectrics which is an achievement but not adequate for a Nature Communication paper in terms of stringent novelty and impact criteria. The paper may be reconsidered for publication in Scientific Report.

Response: We would like to thank the reviewer for the valuable comments and suggestions. We also would like to thank him/her for the seriousness and carefulness in the reviewing process.

Most ferroelectrics undergoing paraelectric-ferroelectric symmetry-breaking phase transitions are usually stimulated by temperature, pressure, or magnetic fields (*Chem. Soc. Rev.* **2016**, *45*, 3811). Light as a unique media can trigger structural changes in

photochromic organics (*Chem. Rev.* **2016**, *116*, 15089). They show the reversible color change, generally resulting from photo-induced geometrical isomerization. Such photo-induced structural change may result in repositioning the orientation of the ferroelectric polarization on the premise of a polar point group in the crystals. The optical control of polarization is a subject of fascination for the scientific community because it involves the establishment of new paradigms for technology. Photoirradiation stands out as a contactless, non-destructive and remotely controlled means beyond electric or strain field. However, although 100 years have been passed since the discovery of the first ferroelectric Rochelle salt, no physicists or chemists have ever studied such optical control of ferroelectricity involving intrinsic structural changes caused by photoisomerization before us. Given this, light-driven isomerized Schiff base ferroelectrics are of milestone significance in the development of ferroelectrics. Recently, our group reported the first photoswitchable ferroelectric crystal, 3,4,5-trifluoro-*N*-(3,5-di-*tert*-butylsalicylidene)aniline, that can achieve polarization switching through a structural phase transition triggered by photoisomerization (*J Am Chem Soc.* **2021**, *143*, 13816). However, its ferroelectric domains are switched slowly under laser irradiation, taking more than one hour to complete the transition to another polarization state, which severely limits its practical applications. Most importantly, the shape and size of its ferroelectric domains are very different in the original and annealed states before and after laser irradiation, and thus the domain structure is difficult to recover.

In this work, we successfully synthesized a pair of single-component organic enantiomorphic ferroelectrics/ferroelastics, (*R*- and (*S*)-*N*-3,5-di-*tert*-butylsalicylidene-1-(1-naphthyl)ethylamine (SA-NPh-(*R*) and SA-NPh-(*S*)), which exhibit not only temperature-dependent thermodynamic ferroelectric/ferroelastic phase transition but also light-driven ferroelectric/ferroelastic phase transition triggered by enol–keto photoisomerization both from $P1$ to $P2_1$ polar space groups. The 2F1-type phase transition is not only a full ferroelectric one but also a full ferroelastic one according to the Aizu rule. Accompanied by the ferroelectric/ferroelastic phase transitions, ferroelectric/ferroelastic domains can be switched quickly within seconds and reversibly between different polarization/strain states. In addition, their spontaneous polarization or ferroelectric domain can also be switched under opposite electric fields in all three phases (room-temperature and high-temperature phases, and photo-triggered phase). To the best of our knowledge, this is the first time that the multi-channel (optical, thermal, and electronic) control of ferroelectric/ferroelastic domains can be realized in organic crystals. These attributes hold promise for multi-channel data storage, on-chip optical interconnects and optoelectronic logic-in-memory devices, and will bring revolutionary changes to smart materials and biomechanical applications in the future.

We agree with the reviewer's point that photo-triggered phase transformation of SA-NPh-(*S*) has been reported. In fact, we are very grateful for this report by Taniguchi *et al.*, and we also cited this paper and mentioned this work in the **Introduction** part of the previous version: "During our systematic search, we found that Taniguchi *et al.* have reported the photo-triggered phase transformation with enol-keto photoisomerization in a homochiral and polar crystal, (*S*)-*N*-3,5-di-*tert*-butylsalicylidene-1-(1-naphthyl)ethylamine (SA-NPh-(*S*)).²⁹" It shows our respect for their work. However, since photochromism was discovered in 1867, the structural phase transition caused by photoisomerization has never been associated with ferroelectricity. That is only the first point. The second point is that in addition to ferroelectric properties, we also combined ferroelasticity with photoinduced structural phase transitions and successfully realized the simultaneous optical control of multiple ferroic orders. Moreover, SA-NPh-(*R*) and (*S*) also exhibit temperature-dependent thermodynamic ferroelectric/ferroelastic phase transition from $P1$ to $P2_1$ polar space groups. Accompanied by the ferroelectric/ferroelastic phase transitions, their ferroelectric/ferroelastic domains can be switched quickly within seconds and reversibly between different polarization/strain states. Consequently, this work realizes the unprecedented multi-channel (optical, thermal, and electronic) control of ferroelectric/ferroelastic domains in organic crystals. However, these points are not the most important feature for SA-NPh-(*R*) and (*S*).

Shape memory alloys have been used extensively in couplings, actuators, medical guide wires and so on, and are hopeful candidates for smart materials that already exist. The unique shape memory effect and superelasticity realized in these alloys are caused by the martensitic phase transformation and its reverse transformation. For ferroic materials, however, almost no memory effects have been found for their ferroic domains after reversible phase transformations. Due to the influence of temperature changing rate and crystal defects, the shape and size of newly emerged ferroic domains often change greatly from the original ones, which can not realize the memory effect. In this work, in addition to the electrically switchable polarization in the ferroic phase, the stripe-like ferroelectric/ferroelastic domain patterns in SA-NPh-(*R*) and (*S*) can completely disappear after heating or UV light irradiation of 365 nm and reversibly reappear after cooling or visible light irradiation of 488 nm. Most importantly, the domain patterns are exactly the same in the initial and final states, demonstrating the existence of a memory effect for the ferroic domains in SA-NPh-(*R*) and (*S*). As far as we are aware, such an intriguing domain memory effect triggered by both thermodynamic and light-driven ferroelectric/ferroelastic phase transformations is the first to be found in ferroic materials. Thermal and optical control of domain memory effect would open up a whole new research field for smart ferroic materials. This is a big innovation and a technological revolution, which would

intrigue chemists, physicists and materials scientists, especially researchers who study photochromic materials. Consequently, we are confident that this manuscript is suitable for *Nature Communications* and our discovery will attract great interest from researchers working in related fields.

The corresponding changes in the revised version:

Shape memory alloys are smart materials that have a broad range of industrial and commercial applications, such as actuators, couplings, medical guide wires, and so on.^{1,2} The reversible martensitic phase transformations bring about the unique shape memory effect and superelasticity in these alloys.³ However, it is difficult for the ferroic domains to show memory effect after ferroics undergo reversible phase transformations. Ferroics, generally including ferromagnets, ferroelectrics, and ferroelastics, have at least two orientation states that could be switched by a certain external field in the low-temperature ferroic phase.^{4,5,6,7,8,9} Taking ferroelectrics as an example, the transformations from ferroelectric phase to paraelectric phase and back to ferroelectric phase would result in increased and broken crystal symmetries.^{10,11,12,13,14,15,16,17,18,19} Due to the influence of temperature changing rate and crystal defects, the shape and size of newly emerged ferroelectric domains often change greatly from the original ones, which can not realize the memory effect. For instance, in two ferroelectric phases of quinuclidinium perrhenate, both domain patterns gained during the cooling treatment are strikingly different from those gained in the original states.²⁰

Enlightened by this pioneering work, we have systematically studied a pair of single-component organic enantiomorphic crystals, SA-NPh-(*R*) and SA-NPh-(*S*), which show not only temperature-dependent thermodynamic ferroelectric/ferroelastic phase transformation but also light-driven ferroelectric/ferroelastic phase transformation triggered by enol–keto photoisomerization both from *P*₁ to *P*₂ polar space groups. Accompanied by the ferroelectric/ferroelastic phase transitions, ferroelectric/ferroelastic domains can be switched quickly within seconds and reversibly between different polarization/strain states. Moreover, their spontaneous polarization or ferroelectric domain can also be switched under opposite electric fields in all three phases (room-temperature and high-temperature phases, and photo-triggered phase). Consequently, ferroelectric/ferroelastic domains can be simultaneously controlled by three different physical channels (light, temperature, and electric field). To our knowledge, this is the first time that the multi-channel (optical, thermal, and electronic) control of ferroelectric/ferroelastic domains can be realized in organic crystals. In addition to the electrically switchable polarization in the ferroic phase, the native ferroelectric/ferroelastic domain patterns (State A) can completely

disappear (State B) by heat or 365 nm UV irradiation and reversibly reemerge (State A') upon cooling or 488 nm visible light irradiation (Fig. 1). Most importantly, the domain patterns in the final state (State A') are exactly the same as in the original state (State A), demonstrating the presence of memory effect for the ferroic domains in SA-NPh-(R) and (S). Such an intriguing domain memory effect triggered by both thermodynamic and light-driven ferroelectric/ferroelastic phase transformations is the first to be found in ferroic materials, which provides a future direction for smart ferroic materials.

Reviewer #3 (Remarks to the Author):

In this manuscript, the authors demonstrate interesting domain memory effects in a pair of single-component enantiomeric organic photochromic multiferroics SA-NPh-(R) and SA-NPh-(S). They simultaneously exhibit unprecedented thermal- and light-driven ferroelectric/ferroelastic phase transitions. More importantly, the ferroic domains can be reversibly controlled by three different physical channels (light, temperature, and electric field) with the perfect reproducibility and memory. This is a landmark work that thermal and optical control of domain memory effect would open up a whole new research field for smart ferroic materials, suggesting the great potential in erasable data storage, optoelectronics, and related applications. Given the manuscript is fluently written, and extensively elaborates the purpose of the research, principal results, and major conclusions, I strongly recommend its publication in Nature Communications after minor revisions.

Response: We sincerely thank the reviewer for spending his/her valuable time on carefully reviewing our manuscript, the in-depth comments, as well as his/her compliments on the impact and significance of our work.

1. The reaction time of photoisomerization in UV, CD, and IR spectroscopy tests in this article is ambiguous. The corresponding reaction time should be given to be more scientifically rigorous.

Response: We thank the reviewer for the good advice. Following the reviewer's suggestion, we have added the reaction time to the corresponding light tests in the measurement methods in the revised Supplementary Information. To monitor the photoreaction process, we have investigated the time-dependent solid-state UV-vis spectrum of SA-NPh polycrystalline sample. As shown in Fig. S24, a new absorption

band around 500 nm gradually becomes saturated within 40s after 365 nm UV irradiation, suggesting that the photoreaction in the surface of the solid sample is basically completed. Subsequently, the new absorption band decreases very quickly and completely vanishes within 10s under further 488 nm irradiation. For CD and IR measurements, to ensure high photoconversion, the irradiation time of both 365 nm and 488 nm lights was up to 60s.

Figure S24. The time-related solid-state UV-vis absorption spectra of SA-NPh-(R) polycrystalline powder sample. a) 365 nm light irradiation. b) photosaturation sample under 488 nm light irradiation.

The corresponding changes in the revised version:

The photoreaction time for UV-vis measurement is 40s under 365 nm UV light and 10s under 488 nm visible light. For CD, VCD and IR spectroscopy tests, the photoreaction time is up to 60s.

2. The authors are suggested to discuss the mechanism of photochromism in detail.

Response: We thank the reviewer for the kind suggestion. As suggested by the reviewer, we have added the mechanism of photochromism in detail in the revised manuscript. Photochromism of organic Schiff base derivatives has been thoroughly investigated since the salicylideneaniline was discovered by Senier et al. in 1909. The photochromic mechanism is based on the photoexcited intramolecular proton transfer of hydroxyl proton and enol-keto structural isomerization, which has been clearly confirmed (Harada *et al.*, *J. Am. Chem. Soc.* **1999**, *121*, 5809–5810) by X-ray crystal structure analysis using two-photon excitation. Thus, from a mechanistic perspective, this provided a specific illustration that photochromism involves the transfer of a proton from the phenolic OH group of the enol form to the N atom of the imine group, generating the *cis*-keto form that is subsequently converted into the red-colored *trans*-keto form. Since the photochromic molecular transformation from enol to *trans*-keto forms occurs without destroying the single-crystal form, it was assumed

that the transformation occurs *via* pedal motion in a manner similar to the thermal motion of azobenzene crystals (Harada *et al.*, *Acta Cryst.* **1997**, *B53*, 662–672; Harada and Ogawa, *Chem. Soc. Rev.* **2009**, *38*, 2244–2252), which is known as a space-efficient motion in crystals.

The corresponding changes in the revised version:

Photochromism in salicylideneaniline Schiff base derivatives involves electron transfer and configuration transformation under the photoexcitation. In detail, when the crystals are irradiated by UV light, the hydrogen proton transfer from the hydroxyl group to the bridge nitrogen atom is activated, and the stable enol form is transformed to the unstable intermediate state of *cis*-keto form. With continuous UV light irradiation, the *cis*-keto form will be immediately transformed to the relatively stable *trans*-keto form with lower steric hindrance that occurs via pedal motion in a manner similar to the thermal motion of azobenzene crystals (Fig. S9).^{32, 33}

3. How about the photo-irradiation effect in P - E hysteresis curve?

Response: We thank the reviewer for the good comment. In fact, we have tried to see what happens to the ferroelectric hysteresis loop after the photoirradiation. However, there is no significant difference between the measured macroscopic hysteresis loops before and after UV light irradiation (Figs. S2d and S8). The main reason might be that unit cell molecular dipoles in Phase I and Phase III adopt similar arrangements, therefore the total polarization of the two phases is similar. Moreover, Schiff base bulk crystals with enol–keto isomerization upon photoirradiation generally have a low conversion rate. For this reason, we carried out PFM measurements to characterize veritable ferroelectric polarization reversal in microscopic, because the surface of thin-film can achieve relatively high conversion efficiency under light excitation.

Figure S8. Polarization–voltage (*P*–*V*) hysteresis loop of SA-NPh-(*R*) by 365 nm UV light in phase III.

The corresponding changes in the revised version:

We also measured the ferroelectric hysteresis loop in Phase III. However, there is no significant difference between the loops in Phases I and III (Fig. S8). The main reason might be attributed to the close total polarization of two phases (Phases I and III). Moreover, the generally low conversion rate of the enol–keto isomerization in the Schiff base crystals upon photoirradiation results in less difference for the ferroelectric loops.

😊 In the end, we sincerely thank the reviewers for their helpful comments and suggestions. By learning those comments, performing new experiments and re-analyzing data, we have gained a lot of in-depth understanding in these materials. To express our sincere appreciation, we have added one sentence in the Acknowledgement part in the manuscript: “The manuscript was improved by the insightful reviews by the anonymous reviewers.”

REVIEWERS' COMMENTS

Reviewer #1 (Remarks to the Author):

The is a good response the referee comments as worthy of acceptance. Much improved.

Reviewer #3 (Remarks to the Author):

The revised manuscript is significantly improved, and I think the current version is acceptable for publication.

Reviewer #1 (Remarks to the Author):

The is a good response the referee comments as worthy of acceptance. Much improved.

Response: We sincerely thank the reviewer for spending his/her valuable time on carefully reviewing our manuscript, the in-depth comments, as well as his/her compliments on the impact and significance of our work.

Reviewer #3 (Remarks to the Author):

The revised manuscript is significantly improved, and I think the current version is acceptable for publication.

Response: We greatly express our appreciation to the reviewer for his/her helpful comments and questions, which are crucial for us to improve the quality of our manuscript. We also would like to thank him/her for the seriousness and carefulness in the reviewing process.